# Early deprivation disruption of associative learning is a developmental pathway to depression and social problems

Margaret A. Sheridan[1], Katie A. McLaughlin [2], Warren Winter[3], Nathan Fox[4], Charles Zeanah[5] & Charles A. Nelson[3,6,7]

Exposure to psychosocial deprivation is associated with elevations in numerous forms of impairment throughout the life-course. Disruptions in associative learning may be a key mechanism through which adversity, particularly psychosocial deprivation, increases risk for impairment. Existing data consistent with this claim come entirely from correlational studies. Here, we present the first experimental evidence relating psychosocial deprivation and disruptions in multiple forms of associative learning. Using data from the Bucharest Early Intervention Project, we demonstrate that randomized placement into a family caregiving environment during the infant/toddler period as compared to prolonged institutional care normalizes two forms of associative learning in early adolescence: reward responsivity and implicit motor learning. These forms of associative learning significantly mediate the effect of institutional rearing on depressive symptoms and peer relationships. In sum, we provide evidence for a novel pathway linking early experience to psychopathology and peer relationships through basic associative learning mechanisms.

[1] Department of Psychology and Neuroscience, University of North Carolina, Chapel Hill, NC 27599, USA. [2] Department of Psychology, Guthrie Hall (GTH), University of Washington, 119A 98195-1525, Seattle, WA 98105, USA. [3] Boston Children's Hospital, Developmental Medicine, Children's Hospital, 300 Longwood Ave, Boston, MA 02115, USA. [4] Department of Human Development, University of Maryland, College Park, MD 20740, USA. [5] Department of Psychiatry and Behavioral Sciences, Tulane University School of Medicine, 1430 Tulane Ave, New Orleans, LA 70112, USA. [6] Department of Pediatrics, Harvard Medical School, 25 Shattuck St, Boston, MA 02115, USA. [7] Harvard Graduate School of Education, 13 Appian Way, Cambridge, MA 02138, USA. Correspondence and requests for materials should be addressed to M.A.S. (email: sheridan.margaret@unc.edu)

Exposure to psychosocial deprivation in the form of institutional rearing is associated with a wide range of adverse long-term outcomes, including high levels of psychopathology[1,2] and poor social functioning[3,4]. We and others have argued that these broad risks are the result of disruptions in basic associative learning mechanisms caused by reduced cognitive and social stimulation in the early caregiving environment[5,6]. These basic learning mechanisms are required to identify patterns and contingencies and predict subsequent environmental inputs[7] conceptualized in recent theories as central to most of aspects of perception and cognition[8]. Evidence supporting these claims is crucial for identifying whether and how early environments shape complex human behavior as well as for identifying potential sensitive periods in development when environmental inputs are most influential[9]. Better understanding of the mechanisms through which early institutionalization confers risk for impairment is also important for identifying novel intervention strategies for children exposed to institutionalization and other forms of adversity. Here, we provide the first experimental test of the hypothesis that basic learning mechanisms are disrupted by exposure to psychosocial deprivation during early childhood.

Deprived environments, such as institutionalization, constrain opportunities for many types of learning early in development. Children raised in institutions lack a stable primary caregiver and spend dramatically less time in the presence of adults than children raised in families[10]. The absence of a caregiver is associated not only with atypical social and emotional development, which are well documented in institutionally reared children[11], but also marked differences in cognitive development[9]. Early in development, interactions with caregivers provide sensory, motoric, linguistic, and social stimulation that foster learning. For example, child-directed speech from the caregiver has been shown to enhance infant attention to environmental stimuli and promote associative learning[12–14]. The absence of a caregiver reduces opportunities for learning and may produce lasting difficulties in associative learning, because opportunities to engage in, and thus practice, these forms learning are unavailable during critical periods of development. We examined how institutional rearing influences two basic forms of associative learning: responsivity to reward and implicit learning (i.e., pattern learning).

Disruptions in associative learning are argued to be a key mechanism linking deprived early environments with impairments later in life[6,15,16]. An increasing body of work suggests that early adversity is associated with differences in some forms of associative learning, particularly reward responsivity[15,17–20], suggesting that early environments may shape these fundamental learning mechanisms. Children exposed to early institutional rearing exhibit reduced learning on a probabilistic reward learning task[21], reduced ventral striatum (VS) activity during reward anticipation on the monetary incentive delay (MID) task that measures behavioral response to reward by iteratively pairing response to a previously neutral stimulus with a reward over time[22], and reduced VS response to positive social images[23]. Disruptions in reward responsivity have also been associated with other forms of early life adversity such as early life stress broadly defined[19], abuse and neglect[18], and sexual abuse[20]. These disruptions in reward processing have been associated with increased risk for depression[23,24] and social skills deficits[21] following exposure to early adversity.

Reward learning is one example of a larger category of basic learning mechanisms that are dependent on interrelated fronto-striatal circuitry[25]. While it is possible that reward learning is specifically disrupted by early adversity exposure, it is more likely that multiple forms of learning reliant on the same neural structures (i.e., dorsal and VS) are also impacted. However, current evidence for this is mixed. One study of children with maltreatment exposure observed associated deficits in a complex probabilistic learning task[15]. However, the only study examining institutional exposure and implicit pattern learning found no relationship[26]. Given the differences in task demands between these studies one possible interpretation is that only specific aspects of implicit learning are associated with early adversity exposure. In typically developing populations, associative learning in infancy predicts later social skills[27] and language acquisition[28,29], suggesting that disruptions in numerous forms of associative learning may be a core mechanism that explains the numerous adverse outcomes associated with early deprivation.

While associations between early environments and reward learning may reflect the impact of early experience on associative learning, it is also possible that these associations reflect shared inheritance. For example, maternal depression has been linked with reward learning in offspring in multiple studies[30–32], an association that could be mediated through genetic or environmental pathways. In the absence of an experimental design, it is impossible to know whether differences in associative learning among children raised in deprived environments are truly the result of experience or simply reflect genetic factors that led parents to place their children in an institution. We present data from a randomized controlled trial of foster care as an alternative to institutional care for abandoned children, the Bucharest Early Intervention Project (BEIP)[33]. We use this experimental data to determine whether randomized placement into family care mitigates the impact of early institutionalization on associative learning and whether improvements in associative learning are a mechanism explaining group differences in mental health and psychosocial outcomes later in development.

In the current study we examine how early life institutional rearing and randomization to a high-quality foster care intervention predict task performance at 12 years of age on a modified MID task[34] (Fig. 1) and on a serial reaction time (SRT) task assessing implicit learning, another basic associative learning mechanism reliant on fronto-striatal systems[35–37]. The use of random assignment allows assessment of the mitigating impact of foster care intervention on the association between early life deprivation and reward and implicit learning. We demonstrate that randomization out of a psychosocially and cognitively deprived institutional environment influences reward learning and implicit learning later in development and that disruptions in these learning processes are mechanisms linking early deprivation to depression and social problems in adolescence. We compliment these analyses by examining the duration of exposure to institutionalization as a predictor of associative learning in adolescence.

## Results
**Behavioral outcomes.** First we examined the impact of group membership on symptoms of depression. We observed a significant main effect of group on depression symptoms ($F_{2,132} = 9.79$, $p < 0.001$). This main effect was due to the fact that children in the FCG and CAUG exhibited more symptoms of depression than children in the NIG ($p < 0.001$). There were no significant differences in symptoms of depression between children in the FCG and CAUG ($p = 0.40$) as we have reported elsewhere[38]. This pattern was identical for caregiver report of depression. Next we assessed the degree to which the amount of exposure to institutionalization was related to symptoms of depression. Among children who had been institutionalized, percent time spent in an institution was not associated with symptoms of depression ($\beta = 0.18$, $t = 1.52$, $p = 0.133$). This association was marginally significant for caregiver report ($p = 0.08$).

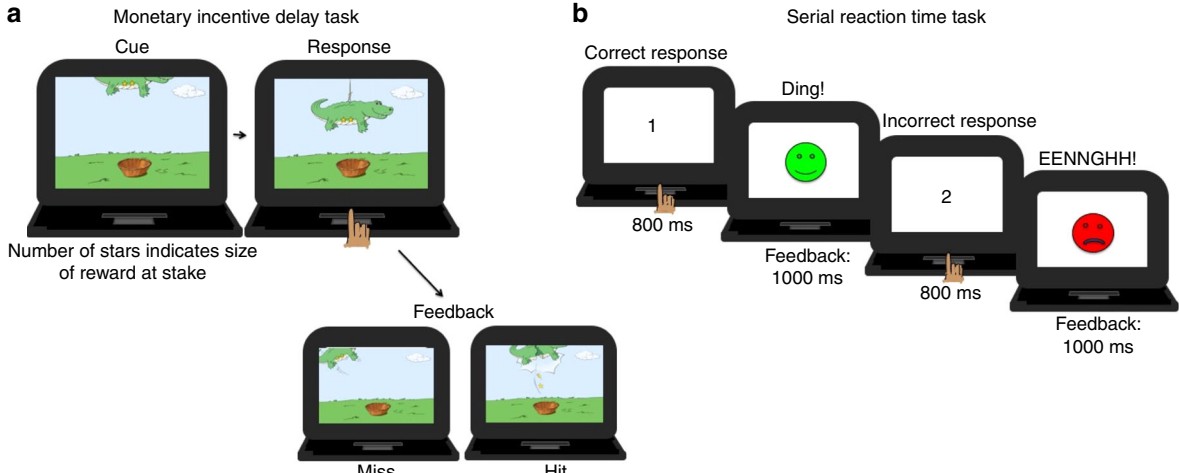

**Fig. 1** Illustration of the MID and SRT tasks. **a** Schematic of the monetary incentive delay task (MID) used in this study showing each phase of a particular trial including presentation of the number of starts a child could earn on a specific trial (screen 1) followed by the child's response (screen 2) and the feedback (hit or miss—screens 3 and 4). Presentation time was modified based on each child's performance on a practice task. Adapted with permission from ref. [34]. All rights reserved. **b** Schematic of the serial reaction time task showing two sample trials. On each trial a number (1–4) appeared for 800 ms, the child simply pressed the button on the keyboard that corresponded with that number. Next feedback occurred—a smiley face and pleasant sound if the keyboard press matched the presented number and a frowny face and unpleasant sound if the press did not match the presented number

**Piñata task**. First we examined main effects of task condition on performance to confirm that we observed patterns of response to the Piñata task consistent with the idea that children are generally learning that their behavior will elicit a reward. Across all groups children performed well above chance on the Piñata Task (average accuracy = 73%) and there was a significant main effect of reward condition on accuracy ($F_{2,133} = 63.7$, $p < 0.001$) indicating that children were shifting their behavior based on the amount of reward available by responding more accurately on the higher reward trials. Across all groups, children were less accurate for 0 star trials compared to 1, 2, or 4 star trials (all $p$'s $< 0.001$).

Next, we identified if there were group differences in the degree to which children increased their accuracy on high vs. low reward trials. There was a main effect of group on overall accuracy ($F_{2,132} = 5.13$, $p = 0.007$) and a significant effect of group on accuracy during 4 star trials controlling for 0 star trial performance ($F_{2,132} = 4.39$, $p = 0.01$), indicating that children in some groups (see below) increased their accuracy on 4 vs. 0 star trials more than others (Fig. 2).

To determine if the impact of group on performance was due to randomization, we performed an intent-to-treat analysis comparing performance for CAUG to the FCG. Consistent with the hypothesis that early environmental experience shapes reward responsivity, we observed that children in the CAUG did not increase their accuracy to reward to the same degree as children in the FCG ($F_{1,87} = 7.25$, $p = 0.009$). The CAUG also did not increase their accuracy to reward to the same degree as NIG ($F_{1,85} = 4.88$, $p = 0.03$); the NIG and FCG did not differ from each other ($F_{1,91} = 0.29$, $p = 0.59$). See Fig. 2.

To identify the degree to which the amount of exposure to institutionalization was related to task performance we examined associations between percent of life spent in an institution and accuracy for 4 relative to 0 star trials. Among children who had been institutionalized, percent time spent in an institution was associated with the degree to which they increased accuracy from 0 to 4 star trials ($\beta = -0.37$, $t = 3.79$, $p < 0.001$).

We also observed a main effect of group on peer relationships ($F_{2,101} = 6.03$, $p = 0.003$). Decomposition of this main effect revealed that there were no significant differences between children in the FCG and CAUG in peer relationships ($F_{2,101} = 0.009$, $p =$ 0.926). However, children in the FCG and CAUG both exhibited worse peer relationships than children in the NIG ($p < 0.001$). Next we assessed whether the amount of exposure to institutionalization was related to peer relationships. Among children who had been institutionalized, percent time spent in an institution was not associated with peer relationships ($\beta = 0.06$, $t = 0.49$, $p = 0.62$). These findings were identical for caregiver report.

Across all participants, reward responsivity significantly predicted symptoms of depression reported by teachers ($\beta = -0.68$, $t = 3.27$, $p = 0.002$). In contrast, reward learning did not significantly predict peer relationships reported by teachers ($\beta = -0.10$, $t = -1.03$, $p = 0.30$). These findings were identical for caregiver report.

Given the strong association between institutional rearing and reward learning, as well as reward learning and depressive symptoms, we examined whether reward learning significantly mediated the effect of intervention on depressive symptoms. We tested this mediation model despite the fact that the intervention effect on depression was null. This approach is in accordance with current recommendations for testing mediation models as a non-significant c-path may result from a particularly strong mediation effect[39–41].

Consistent with our hypotheses, we observed a significant indirect effect of randomized placement into foster care on parent and teacher report of depressive symptoms through reward responsivity (95% bias-corrected confidence intervals (CI): parent [−0.041, −0.003], teacher [−0.061, −0.003]). In addition, reward responsivity mediated the association between percent of life spent in an institution by age 12 and symptoms of depression as rated by teachers (CI: [0.000, 0.003]) but not parents.

**Serial reaction time task**. Across all groups children performed well above chance on the SRT Task (average accuracy = 72%) and there was a significant main effect of condition (pattern vs. random) on accuracy ($F_{2,121} = 5.42$, $p = 0.02$) and RT ($F_{2,121} = 8.56$, $p = 0.004$). Across all groups, children were more accurate and faster on patterned vs. random blocks. These results are as predicted for a SRT task, and are evidence of associative learning in the motor domain (Fig. 3).

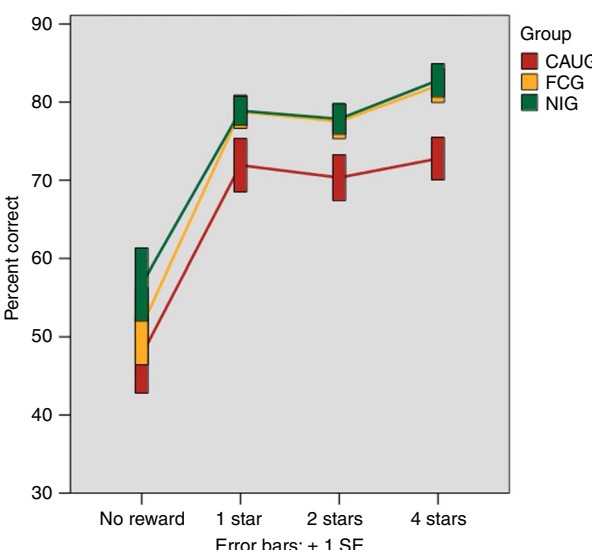

**Fig. 2** Percent correct by group and level of incentive on the monetary incentive delay (Piñata) task. Bars are error bars (±1 SEM; N = 48 FCG; N = 42 CAUG; N = 46 NIG)

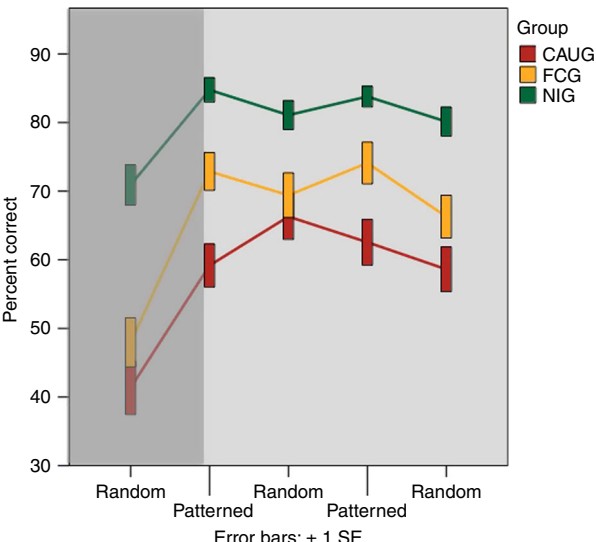

**Fig. 3** Percent correct by group across all blocks of the serial reaction time task. If the block was "random" or "patterned" is indicated on the x-axis. Implicit learning is measured as the degree to which accuracy improves on patterned compared to random blocks The first block (random) is shaded because this block was not included in analyses as performance on this block represents basic learning of the task. Bars are error bars (±1 SEM; N = 44 FCG; N = 37 CAUG; N = 42 NIG)

There was a main effect of group on accuracy ($F_{2,120} = 19.2$, $p < 0.001$) and RT ($F_{2,120} = 9.01$, $p < 0.001$) and a significant effect of group on accuracy and RT during patterned trials controlling for random trial performance (accuracy $F_{2,121} = 9.14$, $p < 0.001$; RT $F_{2,121} = 3.63$, $p = 0.03$), indicating a significant effect of group on the degree to which participants learned the pattern.

To identify if this group effect was the result of randomization to foster care, we directly compared the CAUG and FCG groups. As in the piñata task, we observed a significant intervention effect: children in the CAUG did not learn the pattern to the same degree as children in the FCG as measured by accuracy ($F_{1,80} = 11.26$, $p = 0.001$) but not RT.

Next we examined if the amount of time spent in an institution was related to task performance. Percent time spent in an institution by age 12 significantly predicted accuracy ($\beta = -0.16$, $t = 2.35$, $p = 0.02$) but not RT ($\beta = 0.03$, $t = 0.366$, $p = 0.72$) on patterned relative to random blocks.

Across all participants, pattern learning significantly predicted symptoms of depression reported by teachers ($\beta = -0.72$, $t = 2.27$, $p = 0.03$) but not parents. In contrast to reward learning, pattern learning also significantly predicted peer relationships as reported by teachers ($\beta = 0.49$, $t = 3.14$, $p = 0.002$) and caregivers ($p = 0.01$).

Pattern learning did not mediate symptoms of depression as reported by caregivers or teachers for either the effect of group membership or the percent of life spent in an institution. Next we examined whether pattern learning on the SRT significantly mediated the effect of intervention on peer relationships. Pattern learning, as measured by accuracy, significantly mediated teacher (CI: [0.021, 0.211]; Fig. 4) report of peer relationships but not caregiver report. Similarly, pattern learning significantly mediated the impact of percent of life spent in an institution on peer relationships as measured by teachers (CI:[−0.010, −0.0001]) but not caregivers.

Given the strong relations between depression and peer relationships in adolescence[42,43], we tested the indirect effect of randomized placement into foster care on peer relationships through pattern learning additionally controlling for teacher or parent report of depression. The indirect effect remained significant with these additional controls for teacher report

(CI: [0.012, 0.173]) and became significant for caregiver report (CI: [0.003, 0.119]).

## Discussion

In the current study we demonstrate that prolonged institutional rearing in childhood leads to deficits in reward responsivity and implicit learning observable in early adolescence and that these differences explain later symptoms of depression and difficulties in peer relationships. Specifically, we observe that adolescents who were not randomly assigned out of institutional care performed more poorly on a reward responsivity task and an implicit pattern learning task relative to adolescents who were randomly assigned to high-quality foster care in early childhood. These findings are consistent with recent conceptual models[6,16] and extend observational studies documenting associations between exposure to childhood adversity and blunted neural and behavioral responding during reward tasks in several ways[17,20,22,23]. Here, we provide the first evidence consistent with the interpretation that these associations reflect a causalrelationship, demonstrating a significant difference in reward responsivity as a function of randomization to foster care intervention. Second, we show that this link extends to other forms of associative learning —specifically, implicit motor learning.

Much of the neural structure and function which underlies associative learning, particularly the subcortical components is conserved across species[44] including the central role of dopamine in allowing reward history to guide future behavior[25]. Additionally, disruption in these basic learning mechanisms have been related to a wide variety of negative life outcomes including depression[45], addiction[46], social skill deficits[27], and language acquisition[47] with a remarkable consistency. Given the very basic and primary nature of these learning mechanisms, it is likely that this is a common pathway contributing to multiple areas of impairment following exposure to institutional deprivation. Here we provide novel evidence for this hypothesis, by demonstrating and indirect effect of institutionalization on symptoms of

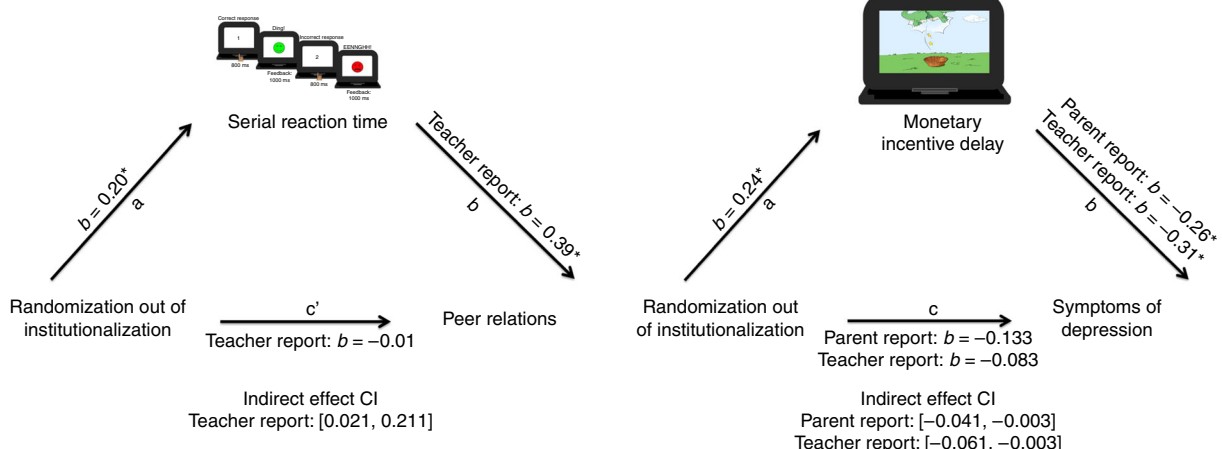

**Fig. 4** Illustration of the primary mediation effects for both SRT and MID tasks. (SRT: $N = 44$ FCG; $N = 37$ CAUG; $N = 42$ NIG; MID: $N = 48$ FCG; $N = 42$ CAUG; $N = 46$ NIG). Confidence intervals are boot-strapped, bias-corrected confidence intervals around the indirect effect. Adapted with permission from ref. [34]. All rights reserved. A * indicates when the a and b paths were significant ($p < 0.05$)

depression and peer relationships through implicit pattern learning and/or reward learning. Interestingly, for implicit pattern learning, mediation of the impact of institutionalization on peer relationships was robust to controls for depression, demonstrating that multiple forms of impairment were independently linked with this basic learning ability.

Why might deprivation influence reward responsivity? One possibility is that children raised in the absence of stable caregiving experience dramatic reductions in the availability of reward in the environment. Early in development, caregivers provide most, if not all, rewards that a child experiences including food, physical proximity, and affection[48]. Shared positive affect exchanges, for example, are associated with secure attachment, and secure attachment is reduced in children who experience deprivation[49–52]. Children raised in institutions receive far less attention and caregiving from adults than children raised in families[10], reducing the frequency with which these rewards are available. A second and related possibility is that this general lack of caregiving specifically deprives children of contingent caregiver responding, which may scaffold early associative learning more generally by producing associations between infant behaviors and reward receipt. Caregivers typically respond contingently to signs of infant distress or even infant "proto-communications" indicating needs (e.g., hunger, discomfort) which do not rise to the level of distress. In response to these indicators, typical caregivers provide physical proximity, soothing behaviors, provision of food, or removal of a source of discomfort/distress[48]. For children raised in institutions, contingent caregiving is dramatically reduced[10], constraining the opportunity to learn that one's behavior can predictably elicit reward from the environment[48]. This absence of early contingent learning may produce lasting changes in multiple forms of associative learning. Consistent with either potential mechanism, we observe here that an absence of species expectant caregiving early in development impacts the degree to which children modify their behavior based on reward properties in early adolescence, extending prior work in observational studies[15,18,21,23,24]. However, we also provide experimental evidence for an effect of early caregiving on implicit pattern learning, indicating that early caregiving disrupts basic associative learning mechanisms, including those that are not driven by reward specifically. Our findings highlight the importance of experimental data, as prior observational work has produced mixed findings about the link between adversity and implicit learning[15,26] Considering that we observe disruption in

multiple forms of associative learning and that these independently mediate complex future behavior, it is possible that previously observed reward responsivity deficits may simply be one instance of a more general disruption in associative learning mechanisms broadly defined. Future research should endeavor to explicate the various forms of frontal–striatal learning that are disrupted by early adversity exposure and their overall impact on a variety of behavioral and health outcomes.

We observed that reward responsivity deficits mediated the impact of institutionalization on symptoms of depression. As we report elsewhere, we did not observe an effect of foster care intervention on symptoms of depression or peer relationships[38]. It may be that this pattern of results reflects a specific impact of foster care intervention on cognitive processing relative to psychological health and well being. This lack of an intervention effect, while concerning, is belied by other evidence from this study showing a long-term impact of early foster care intervention on externalizing psychopathology[1]. Importantly, depression and peer relationships are heterogenous and multi-determined, including by prenatal exposures[53] whereas reward responsivity and implicit motor learning may be more profoundly affected by early postnatal experience.

The pattern of findings here are consistent with a robust literature identifying deficits in reward processing in individuals with major depressive disorder[54,55] including deficits in reward learning[56] and increasing effort to obtain rewards[57] both processes which would lead to poor performance on our MID task. Consistent with our mediation approach, in other populations these deficits occur prior to the onset of symptoms of depression[58]. While the link between associative learning and peer relationships is less well studied, this finding is consistent with previously reported longitudinal associations[27].

We observed that reward learning was entirely remediated by random assignment to a family environment between 6 and 33 months of age; we observe no differences in performance between children who were randomized to foster care or who grew up in a family environment from birth on this task. In contrast, SRT performance was only partially remediated by randomization to foster care, differences in performance on patterned trials between children in the FCG and in the family reared group are still observable even though both groups differ from institutionally reared children. Interpreting the difference in the effect of foster care between these two interventions is complicated by the fact that they are measured in different tasks.

**Table 1 We present data on age, gender, percent of life spent in an institution by participation in the age 12 follow-up visit**

|  | CAUG | FCG | NIG |
|---|---|---|---|
| Age | 12 y 10 m (7.56 m) | 13 y 0 m (7.3 m) | 12 y 11 m (5.2 m) |
| Gender | 44% female | 47.9% female | 57.4% female |
| Percent time in institution[a] | 43.9% (28.4%) | 15% (9.3%) | – |
| Symptoms of depression (Teacher)[b] | 0.47 (0.34) | 0.40 (0.33) | 0.14 (0.25) |
| Symptoms of depression (Parent)[b] | 0.33 (0.30) | 0.25 (0.23) | 0.16 (0.18) |
| Peer relationships (Teacher)[b] | 3.22 (0.65) | 3.23 (0.75) | 3.71 (0.44) |
| Peer relationships (Parent)[b] | 3.19 (0.68) | 3.31 (0.61) | 3.73 (0.35) |

Symptoms of depression rated by parents and teachers on the Health Behavior Questionnaire (HBQ). This is an average rating across 18 items, for each item responses ranged from 0 to 2, more symptoms of depression were indicated by higher average ratings. Quality of peer relationships rated by parents and teachers. This was an average across 19 items, for each item responses ranged from 1 to 4, more positive peer relationships were indicated by higher average ratings
[a] Indicates variables which were significantly different between CAUG and FCG participants
[b] Indicates variables which were significantly different between ever and never-institutionalized participants

However, it maybe that where reward responsivity—by definition a more motivated form of associative learning—can engage multiple neural systems to support task performance, the SRT does not. Thus, group differences in SRT performance may better capture underlying group differences in basic associative learning processes.

The use of the unique BEIP sample limits the number of subjects available and thus our sample size and composition is constrained. Additionally, our implicit learning task provided feedback to participants after each response; as such, the stimuli were not entirely without positive valence and there may be some shared reward processes in both tasks. Relatedly, future research should use tasks which disentangle reward learning from reward responsivity instead of relying on an MID task which taps a number of aspects of reward processing. These changes would allow a more careful comparison of various forms of learning which rely on frontal–striatal systems. Finally, we measure depression and social skills through parent and teacher report using the HBQ which assesses functioning on a number of domains, we hope in future work to measure depression using standardized self report instruments designed specifically to measure psychopathology.

Despite these limitations, this study has unique advantages, we are able to demonstrate in the only dataset of its kind, that prolonged exposure to early life psychosocial deprivation impacts performance on both reward responsivity and implicit motor learning tasks, which in turn mediate symptoms of depression and peer relationships. We provide evidence consistent with this causal pathway in early adolescence. Importantly, this pathway constitutes a marker which is measurable using identical techniques across a wide age range, and thus could be used clinically to identify adolescents at particular risk following early exposure to adverse experiences. Finally, these data are evidence for a new pathway through which early adversity comes to impact a variety of complex behaviors and thus could give rise to novel interventions for families facing adversity.

## Methods

**Participants**. The BEIP is a longitudinal study of children raised from early infancy in institutions in Bucharest, Romania, and is the only randomized controlled trial of foster care as an alternative to institutional rearing ever conducted. A sample of 136 children (aged 6–30 months) was recruited from each of the six institutions for young children in Bucharest. An age-matched sample of 72 community-reared children was recruited from pediatric clinics in Bucharest and comprised the never-institutionalized group (NIG). Between 6 and 33 months of age, in a parallel randomization design, half of children in the institutionalized group were randomized to a foster care intervention, resulting in two groups: the foster care group (FCG) and the group who received care as usual (prolonged institutional care; CAUG). The BEIP study had a policy of non-interference with children randomized to care as usual. Thus, while most of these children remained in institutional care through the age of 5 years, many of them were removed from institutional care at some point and, at the time of this assessment, many live in some kind of family

placement. To measure this variation in care across FCG and CAUG groups we calculated the percent of their lives they had spent in an institution from when they were born up until the current study at ~12 years of age. This variable ranged from 4 to 100% and was significantly different between the FCG and CAUG groups such that children in the FCG spent significantly less time in the institution compared to children in the CAUG ($t(89) = 6.60$, $p < 0.001$). The BEIP was initiated at the request of the Secretary of State for Child Protection in Romania. All study procedures were approved by the local commissions on child protection in Bucharest, the Romanian ministry of health, and the institutional review boards (IRBs) of the home institutions of the three principal investigators (CAN, NAF, CHZ). A more complete description of procedures employed to ensure ethical integrity has been published previously and commented on by the scientific community[59,60].

Approximately 12 years after the formal RCT was initiated, we acquired task data from 138 participants (N = 48 FCG; N = 43 CAUG; N = 47 NIG) in early adolescence (49.6% female; 11.14–14.62 years of age). Of these, two were unable to complete either task because of fatigue (N = 1, NIG) or because of technical difficulties (N = 1, CAUG). Data from an additional 12 children was not available from the implicit learning task (5 CAUG, 4 FCG, and 4 NIG) due to a technical problem with data collection. Thus our final sample was N = 136 (MID) and N = 124 (SRT). No significant differences were found between the CAUG, FCG, and NIG in age (p = 0.67) or gender distribution (p = 0.43). See Table 1.

**Piñata task**. The Piñata Task is a child-friendly version of the MID task used in children as young as 5 years to assess sensitivity to reward sensitivity and reward learning[34] (Fig. 1a). While the MID was not originally designed as a reward learning task, in other work we have demonstrated that this class of tasks yields learned associations with the stimuli which predict reward which subsequently affects behavior[61]. Here we refer to this task as measuring 'reward responsivity' to indicate that is measures a number of constructs related to reward response. Participants were instructed to collect as many points as possible during the task by hitting a series of "piñatas" shaped like cartoon animals in order to win $10 at the end of the game. On each trial participants saw part of the piñata in the shape of an animal that had 0, 1, 2, or 4 stars on its belly. These stars indicated the number of points a participant could win on each trial and functioned as the "cue". Next the piñata moved into the "strike zone" and participants were instructed to press the space bar when they could see the whole animal. If the participant pressed the button rapidly enough when the piñata was in the strike zone, the piñata broke open and the stars fell into a waiting basket, indicating that the participant earned those "points". If the participant did not press rapidly enough, the piñata swung away, unbroken, with the stars still in its belly indicating that the participant did not earn those points. If the participant pressed before the piñata entered into the strike zone it did not break open, indicating that they did not earn those points.

Prior to the task, each participant played a "practice game"—a 22 trial calibration phase—during which average reaction time (RT) was calculated. After this phase was completed, the display time for the target was no longer calibrated based on performance during the task itself, allowing participants to improve on their own performance throughout the task. Average RT was used during the subsequent task in the following way: the piñata was displayed in the strike zone for a pre-specified range of durations calibrated based on the participant's observed RT during the practice (e.g., it was displayed for 500, 550, or 600 ms). A range of times were used to decrease the degree to which the participant could predict when the piñata would move into the strike zone. Variables of interest on this task include accuracy on 0, 1, 2, and 4 star trials. Reward learning is inferred by comparing accuracy on 0 point trials to accuracy on 4 point trials. If children adjusted their behavior in response to the amount of reward available across the task, they should be more accurate on 4 star trials than when no reward is available.

**Implicit learning task**. Implicit learning was assessed using a SRT task[62]. Participants were asked to press one of four buttons on a button box labeled 1, 2, 3, or 4 that corresponded with a number presented on the screen (Fig. 1b). Numbers were

presented in black in the center of a white screen, and all participants sat in the same spot relative to the screen. On each trial, numbers were presented and responses recorded for 800 ms. To enhance attention to the task, each trial was followed by feedback indicating to the participant if they had pressed the correct button (green smiling face & pleasant sound if they pressed, for example, the "1" button when the 1 was presented), the wrong button (red frowning face and unpleasant sound if they pressed, for example, the "2" button when the 1 was presented), or if they did not respond quickly enough ("Too slow!" if they responded outside the 800 ms response window). Feedback was presented for 1000 ms.

Performance was divided into five blocks. During the first, third and fifth blocks participants saw the numbers 1–4 presented in a random order. During the second and fourth blocks, participants saw the following 10-item pattern repeated five times: 3 1 2 3 4 2 1 3 2 4. Implicit learning is inferred by comparing average RT and accuracy during blocks 3 and 5 (random blocks) and blocks 2 and 4 (patterned blocks).

**Depressive symptoms and peer relationships**. At age 12, we administered the McArthur Health and Behavior Questionnaire (HBQ) to each child's caregiver and teacher to assess symptoms of depression and social functioning[63]. The HBQ is a well-validated assessment of symptoms of psychopathology and child functioning and is sensitive to the presence of internalizing psychopathology[64]. The HBQ was translated into Romanian, back-translated into English, and assessed for meaning at each step by bilingual research staff. For depression, we used the 18-item sub-scale that assesses symptoms of depression. For peer relationships we summed across three HBQ subscales: peer acceptance/rejection (8 items), relational victimization (6 items) and bullying (5 items). HBQ data was unavailable for 1 caregiver and 31 teachers. Because of group differences in the duration of exposure to and relationship with caregivers we primarily report on teacher assessment of depression and peer relationships.

**Statistical analysis**. For all analyses we follow intent to treat, thus group differences between CAUG and FCG are interpreted as consistent with the causal impact of randomization out of a deprived environment on behavior. We identify group differences in task performance and the main effect of task using ANOVA. Conditions of interest are group (CAUG, FCG, NIG), condition (0 or 4 stars or random/pattern). To identify treatment effects on task conditions of interest we ran an ANOVA comparing performance for CAUG to FCG on 4 vs. 0 star trials or pattern vs. random trials. We follow these analyses by comparing both FCG and CAUG to NIG. To examine the association between symptoms of depression, and task performance we used OLS regression. All analyses controlled for age and gender. In Table 1 we include means and standard deviations for each of our dependent variables by group.

In all mediation analyses we tested the direct comparison of FCG to CAUG with NIG serving as the omitted group or baseline using unweighted effect codes[65]. This approach includes two covariates first comparing FCG to CAUG and then a control covariate comparing both FCG and CAUG to NIG. We interpret the results from the first covariate in these analyses and test mediation for the experimental effect of foster care intervention on depressive symptoms and peer relationships. First, we examined the total effect of foster care intervention on symptoms of depression/peer relationships (C path). We examined associations between institutional rearing and task performance (MID, SRT) (A path), and next, associations between task performance (MID, SRT) and psychopathology (B path). If associations were significant for a single path, we tested the indirect effect[66–68]. Boot-strapped, bias-corrected CIs were estimated (5000 resamples) for the indirect effect, which is appropriate for small samples and non-normality in the standard errors of indirect effects[68].

**Data availability**. These data are not currently available for public use as they contain information that could potentially compromise research participant privacy. Please direct any communications concerning data availability to the corresponding author.

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

## Acknowledgements

This research was supported by a grant from the John D. and Catherine T. MacArthur Foundation Research Network on Early Experience and Brain Development (Charles A. Nelson, Network Chair) and the National Institutes of Health (R01-MH091363 to Nelson, K01-MH092526 to McLaughlin, and K01-MH092555 and R03-DA037405 to Sheridan). These funders provided support for all data collection and analysis. We thank the caregivers and children who participated in this project; and the Bucharest Early Intervention Project staff for their tireless work on our behalf.

## Author contributions

M.A.S. and C.A.N. had full access to all of the data in the study and takes responsibility for the integrity of the data and the accuracy of the data analysis. Study concept and design was done by M.A.S., K.A.M., C.A.N., C.Z., and N.F. Statistical analysis was performed by M.A.S., and W.W. Analysis and interpretation of data was performed by M.A.S., K.A.M., C.A.N., C.Z., and N.F. M.A.S. drafted the manuscript. M.A.S., W.W., K. A.M., C.A.N., C.Z., and N.F. critically revised the manuscript for important intellectual content. Study was supervised by M.A.S., K.A.M., C.A.N., C.Z., and N.F.

## Additional information

**Competing interests:** The authors declare no competing interests.

