## [Peer Review File · Nature Communications]

Reviewers' comments:

Reviewer #1 (Remarks to the Author):

This manuscript presents a study examining the effect of institutional rearing on associative learning processes. Children approximately twelve years of age completed a reward task and a serial reaction time task. Performance on both tasks was improved in children who were randomly assigned to foster care, versus those who received continued institutional care, who remained in institutional care. The study provides strong causal evidence that deficits in cognitive processes as a result of institutional rearing are ameliorated by placement into family care. These findings are novel and important, as they clearly indicate the benefits of the family rearing environment for normative cognitive development.

I have a number of minor comments, questions, and suggestions, as well as one primary concern about the analyses of the relation between learning and psychopathology and social skills, which calls into question some aspects of the framing of the paper and the interpretation of the findings.

First, the minor comments:

- The MID is described here as a reward learning task. However the task does not require learning of associations between cues or actions and rewarding outcomes, but instead requires effort to rapidly and precisely execute of motor responses to successfully obtain a rewarding outcome. Many studies in healthy adults fail to observe accuracy or RT differences across reward conditions in this task, and the poor performance in CAUG (reflected by an absence of improved motor performance for trials carrying with higher incentives) could just as easily stem from dysfunctions in motor control as from associative learning difficulties. This might be mentioned as a caveat in the introduction of the task, or in the discussion.

- While the details of this study population have been described elsewhere, it would be helpful for the reader to include a little more detail about the duration of institutionalization for the CAUG group, and the current living environments at the time of testing for both the FCG and CAUG. These participants were tested in early adolescence, but there is no information about the amount of time that participants the CAUG remained in institutional care, or that the FCG remained in foster care.

- Related to this last point, it would be helpful to present analyses of whether duration of institutionalization has similar explanatory power as the randomized group assignment in these cognitive effects?

- The paragraph describing the calibration of the time that the piñata was displayed in the strike zone is confusing. A description of how the display duration was calculated as a function of RT would be helpful (e.g., normally distributed around the mean RT with a specific SD?). The paragraph goes on to state "once the real game had commenced, display time was no longer manipulated". Does this mean that display times in the strike zone were

fixed during the task (if so, how long) and only variable during the practice game? Greater clarity here would be helpful.

My primary concern lies with the mediation result. The mediation analysis is presented as providing evidence that deficits in associative learning contribute to depressive symptoms and poor social skills and are ameliorated by foster care. However the mediation analysis does not appear to provide clear support for this interpretation.

First, the coding strategy (CAUG: -1, NIG: 0, FCG: 1) used for the mediation is confusing. If the goal is to contrast the effect of foster care versus institutionalization, the data from the controls could simply be omitted, but it is unclear why the NIG would be treated as a baseline group with FCG effects assumed to be higher than NIG, and CAUG assumed to be lower. The rank ordering of performance results in the study tasks had the NIG group performing better than (SRT) or as well as (MID) the FCG group. Statistics for the mediation a-path are never provided, but it is unclear to interpret the effect that this path would capture with the groups coded as described. It would be preferable to see a continuous variable such as duration of institutionalization used in the mediation analysis instead of a categorical grouping that encodes assumptions about the directionality of the anticipated effects.

The relationship between task performance (on each task) and depressive symptoms and social skills (the b-path of the mediation) is reported as significant at the full group level, but the significance or directionality of these relationships in each subgroup need to be reported as well in order to interpret the mediation analysis. The CAUG and FCG differed in their performance on the MID and SRT, but did not differ in terms of their ratings for depressive symptoms and social skills.

Thus, I can imagine two patterns of b-path effects that enable this significant mediation must be either 1) the improvement in cognitive performance in the FCG group relative to the CAUG (a-path) is neutralized by an oppositely signed relationship in the predictive power of these cognitive abilities for depressive symptoms and social skills (b-path), (i.e. lower in FCG relative to CAUG), or 2) the full group correlation reported between MID and SRT performance and depressive symptoms and social skills is driven by the NIG participants, with a null relationship for the CAUG and FCG groups that negates the influence of their differences in associative learning on their depressive symptoms and social skills.

In either case, this mediation would not support the interpretation that improvements in associative learning produced by foster care also ameliorate deficits in social skills or increases in depressive symptoms, but instead would suggest dissociation between the social and cognitive effects of institutionalization. Whereas the cognitive deficits associated with institutionalization are ameliorated by foster care, the adverse socio-emotional consequences appear not to be. If this is the parsimonious conclusion to be drawn from the data, it is still a very interesting and novel finding, but would require some adjustment in framing of the current manuscript.

It would be helpful to provide greater clarity about the pattern of associations within each group that produce the mediation effect described in the manuscript, and to consider whether these patterns would be better accounted for by the “dissociation” interpretation presented above.

Reviewer #2 (Remarks to the Author):

This is a truly remarkable study, the challenges of which should not be understated. The randomization is a considerable strength. I think the data are of considerable interest to a wide readership, although probably not for the reasons stated by the authors. The comments below expand on these issues, but in brief, the study is not a direct examination of the “causal” effect of extreme social deprivation on later cognitive-emotional function, but rather a study of the effects of an intervention designed to provide a wide range of experience denied in early life.

I am somewhat puzzled why the authors have focused more on the intention of defining the effects of early social deprivation and less so on the effects (or lack of) the intervention? Also, it is sobering that despite the rather robust effects of the foster care treatment on experimental tasks, there was apparently little effect on emotional well being or social function, at least as rated by teachers/caretakers. Does this speak to the issue of reversibility?

I think both the introduction and especially the Discussion are over-written, and could be shortened considerably, with an emphasis on the more issues noted above. Finally, I think the experimental tests, while appropriate, do not inform on the specific neural processes at play.

In sum, I would very much suggest a considerable re-writing that respects the actual design of the study (an intervention study that only indirectly informs on the impact of social deprivation) and the actual outcomes, or lack thereof, the intervention on socio-emotional function. The intense deprivation associated with institutionalized rearing is accompanied by multiple effects not likely reversed. For example, I recall data on growth stunting in these children as well as altered pubertal timing. Is there data on the general health of the groups (e.g., infection rates. Etc)? I think a more focused discussion of the reversible and, apparently, non-reversible effects is of relevance in relation to the multiple features of the institutional setting. Moreover, if I understand, the placement of children in these institutions was non-random, thus imply adversity at earlier periods of development, including the prenatal period. Interestingly, Pearson et al (JAMA Psychiatry 2013) suggest that the transmission of risk from depressed mother to offspring is primarily during prenatal development – could this explain the negative findings with depressive symptoms? I believe the paper could broaden what is currently a rather narrow focus to discuss the negative findings on depressive symptoms and social skills.

Comments

p. 2: The authors claim that "The absence of a caregiver results not only in atypical social and emotional development, ... but also marked differences in cognitive development.⁸" This statement implies causality and seems at odds with the gist of the abstract, which rightly notes the limitation of previous studies as correlational.

p. 3: The phrase "other forms of early-life adversity such as early life stress broadly defined" is rather meaningless – why not simply "other forms of early life adversity"? Presumably early life adversity and early life stress are somewhat synonymous. I am also unclear as to what is meant by "disrupted parenting" – this should be more precise, such as in the other examples.

p. 3: I am also confused about the source of the evidence for the statement "These disruptions in reward processing have been associated with increased risk for depression^{22,23} and social skills deficits²⁰ following exposure to early adversity." Is there mediational analyses? Deficits in reward processing are indeed associated with a risk for depression, but this is not unique to individuals experiencing early life adversity.

p. 3: RDoC is a working model, not a source of evidence.

p. 4: I think the authors are actually doing a disservice to their work. The model does not actually address the issue the "causal relationship between early life adversity with reward and implicit learning" rather it speaks to a more specific form of adversity, that of the lack of age-appropriate social interactions with adults. It does not address the issue of 'early life adversity', which can take many forms. I think this is a critical point as this literature tends to very wrongfully lump multiple, very diverse forms of experience as "adversity", which does a real disservice to the developmental science. Finally, an experiment that address the causal role of an adversity would randomly assign individual to a condition of that adversity.

p. 4: Regarding the "An age-matched sample of 72 community-reared children was recruited from pediatric clinics..." were these children seen for specific disorders/treatments, or routine pediatric examinations? I think this needs details. Also, was there analysis of the home environments or parent-offspring interactions in the foster care settings? This is actually the intervention, and yet is poorly described.

p. 4: Were the 6 institutions of comparable care (sic)? Are there systematic differences between children placed in institutions and those that remain within their families? This bears on the later FCG and CAUG to NIG comparisons.

p. 4: Can the authors elaborate on "care as usual (prolonged institutional care; CAUG)."? This seems to be the critical comparison group.

p. 7: I do not really understand how group differences are interpreted as "the effect of early caregiving environment". This is not what was manipulated and seems counter to the

design, which is an intervention study. Why would group differences not be associated with the placement in foster care? They were not randomly assigned to adverse conditions?

"Consistent with our hypotheses, we observed a significant indirect effect of randomized placement into foster care on parent (95% bias corrected confidence interval (CI): [.003, .072]) and teacher (CI: [.004, .042]) report of depressive symptoms through reward learning." As noted below, reward sensitivity is a defining feature of depressive states, so this finding is not surprising. This link would be strengthened by reference to high-risk populations (and see below). Are there data on drug use in these samples (e.g., cigarette use)? I am also somewhat surprised by the focus on depression and an additional assessment of anxiety-related symptoms, which normally show an earlier onset.

p. 12: The authors "examined whether pattern learning on the SRT significantly mediated the effect of intervention (CAUG vs. FCG) on depressive symptoms." But on p. 10 the authors state that "There were no significant differences in symptoms of depression between children in the FCG and CAUG (p's .16, .40)...", suggesting no intervention effect.

p. 12: I think the lead statement in the Discussion, for reasons also noted above, overstates (misstates) the results "In the current study we demonstrate that early institutional rearing causes deficits in reward and implicit learning observable in early adolescence and that these differences explain later impairments in both depression and social skills. ". The study directly examined the effects of an intervention, that included dimensions associated with institutionalization – this is well off the claim of evidence for a causal relation between institutionalization and the performance differences in the specific tasks. Moreover, there were no differences in depressive symptoms between the FCG and CUAG groups, thus contradicting this statement. Likewise on p. 10 the authors acknowledge that there were "no significant differences between children in the FCG and CAUG in social skills." The merits of the mediational models, which appears driven with the addition of the NIG group, is that the deficits in the reward and implicit learning tasks appear to be associated with the behavioral problems – however, this remains based on correlational models.

p. 12: I do not see how this is a ", implicit motor learning" – what is the motor learning component?

p. 13: I think comparative anatomists would argue against the notion that "neural structure and function which underlies associative learning is conserved across species.." – the incentive components, perhaps, but the neocortical components – not so much.

That aside, the authors have used appropriate tests for forms of learning – the nature of the impaired performance, however, remains to be determined. There are, as I am sure they appreciate, attentional mechanisms, amongst other functions, that contribute to performance. And the absence of a negative control (i.e., a learning task on which the FCG/CAUG and NIG groups do not differ) limits interpretation.

p. 14: There is a circularity in statements that associated reward learning to depression...anhedonia is a defining feature, perhaps the defining feature, of depression.

Hence by definition reward processing is associated with depression. The studies of Ian Gotlib with high-risk subjects may help address this point. Unlike the current paper, the model cited shows specific processing deficits and neural circuits derived from fMRI analyses.

p. 15: "We observe that reward learning was entirely remediated by random assignment to a family environment between 6 and 33 months of age; we observe no differences in performance between children who were randomized to foster care or who grew up in a family environment from birth on this task (Figure 2)." This is indeed perhaps the strongest finding. One issue is that of decomposing the performance deficits. Is it relevant that the groups differed very significantly on the 'random' condition in the implicit learning task? Would I be correct to assume that this would seem to be a purer test of memory – might this speak to the issue of specificity? What does it say about the children in general?

One the same point, I am somewhat puzzled by the term "reward learning"? What is actually learned here? This test appears to examine reward-sensitive performance, which is not trivial. But I do not see the actual "learning" component?

Reviewer #3 (Remarks to the Author):

This manuscript examined associations between early deprivation, reward learning and implicit learning, and depressive symptoms and social skills. It is the first experimental (non-correlational) study to report that associative learning mediates associations between early deprivation and adolescent behavioral outcomes. Researchers in the field have speculated that early stress influences mental health and social skills via associative learning, and it is exciting to see preliminary experimental evidence for this relationship. The statistical analyses appear to be sound and the manuscript is very well-written. I only have a few concerns, which are listed below.

The measurement of depression (7 items) is limited. It is good that the authors obtained both parent and teacher reports, but an additional child report and a more detailed measure like the CDI would make the findings more robust. The limited measurement of depressive symptoms should be discussed.

For social skills, the items used seem to measure primarily peer relationships, but not broader social skills. The authors might consider using the term "peer relationships" to describe this construct.

It would be useful to see a graph or table showing the distributions of scores for depression and social skills in each group.

Was pubertal development measured in this study? Puberty is often more advanced in children exposed to early stress, and is also related to depression and reward learning.

Did any children fail to perform above chance on the MID and SRT, and were they eliminated from data analysis?

It should be noted in limitations that this study was not completely experimental, given the inclusion of the never-institutionalized group.

Reviewer #1 (Remarks to the Author):

This manuscript presents a study examining the effect of institutional rearing on associative learning processes. Children approximately twelve years of age completed a reward task and a serial reaction time task. Performance on both tasks was improved in children who were randomly assigned to foster care, versus those who received continued institutional care, who remained in institutional care. The study provides strong causal evidence that deficits in cognitive processes as a result of institutional rearing are ameliorated by placement into family care. These findings are novel and important, as they clearly indicate the benefits of the family rearing environment for normative cognitive development.

I have a number of minor comments, questions, and suggestions, as well as one primary concern about the analyses of the relation between learning and psychopathology and social skills, which calls into question some aspects of the framing of the paper and the interpretation of the findings.

We thank the reviewer for their positive overall evaluation and address these concerns below.

First, the minor comments:

- The MID is described here as a reward learning task. However the task does not require learning of associations between cues or actions and rewarding outcomes, but instead requires effort to rapidly and precisely execute of motor responses to successfully obtain a rewarding outcome. Many studies in healthy adults fail to observe accuracy or RT differences across reward conditions in this task, and the poor performance in CAUG (reflected by an absence of improved motor performance for trials carrying with higher incentives) could just as easily stem from dysfunctions in motor control as from associative learning difficulties. This might be mentioned as a caveat in the introduction of the task, or in the discussion.

We have mentioned this as a caveat in the discussion and adjusted our language around this task description and in our interpretation. While the MID was not originally designed as a reward learning task, in other work we have demonstrated that this class

of tasks yields learned associations with the stimuli which predict reward which subsequently affects behavior (Winter & Sheridan, 2014). Specifically, cues associated with reward elicit a different pattern of behavioral response following the task than cues that were not associated with reward, indicating learning of a reward-cue association. However, here we do not report on the effect of this learning but instead on our calculation of the change in response time to high-reward versus non-rewarded trials. As the reviewer states, this construct reflects the degree to which a child modulates their behavior in response to reward value. This measure has frequently been used in both child and adult studies of reward processing (Pizzagalli et al., 2009; van Hulst et al., 2015). Further, because we used a staircase procedure to adjust presentation time of the stimuli only during practice, and then presented stimuli at a set range of durations during the task, it is appropriate to measure reaction time as an index of the degree to which participants were able to move their behavior to reward over time. We explain these procedures more clearly in the methods section in response to this and the below comment asking for better explanation of the procedures followed in this task.

While the details of this study population have been described elsewhere, it would be helpful for the reader to include a little more detail about the duration of institutionalization for the CAUG group, and the current living environments at the time of testing for both the FCG and CAUG. These participants were tested in early adolescence, but there is no information about the amount of time that participants the CAUG remained in institutional care, or that the FCG remained in foster care.

- Related to this last point, it would be helpful to present analyses of whether duration of institutionalization has similar explanatory power as the randomized group assignment in these cognitive effects?

We have included more of this information in the Participants section. Children in the FCG group were randomly assigned out of institutional care between 6-36 months of age. However, as has been detailed in prior papers (Nelson et al., 2007; Zeanah et al., 2003), the study had a policy of non-interference with children randomized to care as usual. Specifically, the study played no role in determining the placement of these

children following randomization. Most of these children remained in institutional care through the age of 5 years, but many of them were removed from institutional care at some point and, at the time of this assessment, many live in some kind of family placement.

Because there is variation in the duration of time spent in the institution across both the FCG and CAUG groups, duration of institutional care can be used as an indicator of the extent or severity of exposure as a compliment to looking at randomization group. We now report on this variable in the manuscript. Importantly, we also report that this variable is strongly associated with group assignment, as one would expect, such that children in the FCG spent significantly less time in the institution compared to children in the CAUG ($t(89) = 6.60, p < .001$). Because the study was designed with randomization and an intent-to-treat analysis is the most appropriate and conservative approach and allows causal inference we continue to report on the intent-to-treat effects. Given that many readers, like this reviewer, might be interested in observing the association between our outcomes and the severity or extent of exposure, we also report on percent time in the institution in the manuscript. For both the MID and SRT, we observe that the percent of your life spent in the institution by age 12 was associated with task performance, such that increased time spent in the institution predicted poorer performance (see pages 10 and 13).

- The paragraph describing the calibration of the time that the piñata was displayed in the strike zone is confusing. A description of how the display duration was calculated as a function of RT would be helpful (e.g., normally distributed around the mean RT with a specific SD?). The paragraph goes on to state "once the real game had commenced, display time was no longer manipulated". Does this mean that display times in the strike zone were fixed during the task (if so, how long) and only variable during the practice game? Greater clarity here would be helpful.

We've added additional details to make the procedures more transparent here. The purpose of the practice trials here was to calibrate the speed of the task (i.e., how quickly each participant needed to answer in order to get a trial correct). This was done to ensure that the task was equally difficult for each participant, which was an important consideration to ensure that the task was challenging and engaging for

participants across groups and to adjust for individual differences in reaction time (which are fairly substantial across groups). After the 22 trial calibration phase was completed, the display time for the target was no longer calibrated based on performance during the task itself, allowing participants to improve on their own performance throughout the task—we highlight this aspect of the task more clearly in the task description. Thus variability in the measurement of total stars meaningfully reflects individual differences in reward performance.

My primary concern lies with the mediation result. The mediation analysis is presented as providing evidence that deficits in associative learning contribute to depressive symptoms and poor social skills and are ameliorated by foster care. However the mediation analysis does not appear to provide clear support for this interpretation.

First, the coding strategy (CAUG: -1, NIG: 0, FCG: 1) used for the mediation is confusing. If the goal is to contrast the effect of foster care versus institutionalization, the data from the controls could simply be omitted, but it is unclear why the NIG would be treated as a baseline group with FCG effects assumed to be higher than NIG, and CAUG assumed to be lower. The rank ordering of performance results in the study tasks had the NIG group performing better than (SRT) or as well as (MID) the FCG group.

I think some confusion exists here with regards to the regression design. Using a covariate like the one we describe does not rank order the groups, rather it directly compares the FCG and CAUG group while including the NIG as a baseline, which is appropriate because the NIG were raised in typical home environments and conceptually represent our baseline condition. This approach is described as using unweighted effect codes and is the appropriate statistical approach given the questions asked here (Cohen, Cohen, West, & Aiken, 2003). Here we coded NIG as 0, CAUG as -1, and FCG as 1. A further control comparing both EIG groups to NIG (e.g., 1, -.5, -.5) can be added to this regression design. When we include this control the same pattern of results is largely unchanged so we now report those findings.

Statistics for the mediation a-path are never provided, but it is unclear to interpret the effect that this path would capture with the groups coded as described. It would be preferable to see a continuous variable such as duration of institutionalization used in the mediation

analysis instead of a categorical grouping that encodes assumptions about the directionality of the anticipated effects.

Statistics for the a-path were provided in the section titled “Task Performance” under each Task heading. As stated above, this coding does not include assumptions about the directionality of effects. We agree completing all analyses with duration of institutionalization would be interesting, and we provide those results below and in the revised manuscript. Importantly, this analysis should be considered supplementary, as it does not rely on the randomization and intent-to-treat approach. To examine duration of institutionalization we completed each analysis with %time institutionalized at age 12 as our predictor (this variable is only available for the FCG and CAUG groups).

Percent time in the institution at age 12 predicted performance on the MID and SRT in the expected direction, such that more time spent in the institution was associated with relatively poorer performance on high versus low reward trials (MID) and patterned versus random trials (SRT). When percent time in the institution was used as a predictor in the mediation analyses the impact of time spent in the institution on depression was mediated by MID performance and the impact of time spent in the institution on social skills was mediated by SRT performance. These results are now included in the manuscript.

The relationship between task performance (on each task) and depressive symptoms and social skills (the b-path of the mediation) is reported as significant at the full group level, but the significance or directionality of these relationships in each subgroup need to be reported as well in order to interpret the mediation analysis. The CAUG and FCG differed in their performance on the MID and SRT, but did not differ in terms of their ratings for depressive symptoms and social skills.

Reporting associations between the mediator and dependent variable by level of the independent variable is non-standard and somewhat problematic because each of those comparisons has significantly reduced power relative to the overall model, thus we are concerned about including it in the paper. However, we ran some of these analyses to address the concern that is stated below and we include that information here.

1) the improvement in cognitive performance in the FCG group relative to the CAUG (a-path) is neutralized by an oppositely signed relationship in the predictive power of these cognitive abilities for depressive symptoms and social skills (b-path), (i.e. lower in FCG relative to CAUG).

This doesn't appear to account for our findings. For example, the association between social skills and SRT performance was ($b=.81$ $t=2.59$ $p=.02$) for the CAUG, ($b=.31$ $t=1.28$ $p=.2$) for the FCG, and ($b=.23$ $t=.86$ $p=.39$) for the NIG. For depression the association between depression and MID performance was ($b=-.19$ $t=-1.1$ $p=.28$) for the CAUG, ($b=-.25$ $t=-1.8$ $p=.09$) for the FCG, and ($b=-.29$ $t=-1.9$ $p=.07$) for the NIG. Again, as the reviewer can see, these associations are all consistently in the same direction and while the significance differs slightly across groups due to variations in sample size.

2) the full group correlation reported between MID and SRT performance and depressive symptoms and social skills is driven by the NIG participants, with a null relationship for the CAUG and FCG groups that negates the influence of their differences in associative learning on their depressive symptoms and social skills.

This also does not appear to account for our findings. There were 5 significant b-path associations reported in the paper. For only two of these associations was the standardized beta value for the association in the NIG group larger than both the CAUG and FCG groups.

Social Skills	CAUG	FCG	NIG
SRT (Parent)	$b=-.03$ $t=-.08$ $p=.93$	$b=.58$ $t=2.43$ $p=.02$	$b=.21$ $t=.96$ $p=.34$
SRT (Teacher)	$b=.81$ $t=2.59$ $p=.02$	$b=.31$ $t=1.28$ $p=.2$	$b=.23$ $t=.86$ $p=.39$

Depression	CAUG	FCG	NIG
MID (Parent)	$b=-.19$ $t=-1.1$ $p=.28$	$b=-.25$ $t=-1.8$ $p=.09$	$b=-.29$ $t=-1.9$ $p=.07$
MID (Teacher)	$b=-.38$ $t=-2.2$ $p=.04$	$b=-.07$ $t=-.40$ $p=.69$	$b=-.43$ $t=-2.5$ $p=.02$

SRT (Teacher)	$b=.15$ $t=.45$ $p=.65$	$b=-.38$ $t=-1.45$ $p=.16$	$b=-.33$ $t=-1.3$ $p=.21$
---------------	-------------------------	----------------------------	---------------------------

In either case, this mediation would not support the interpretation that improvements in associative learning produced by foster care also ameliorate deficits in social skills or increases in depressive symptoms, but instead would suggest dissociation between the social and cognitive effects of institutionalization. Whereas the cognitive deficits associated with institutionalization are ameliorated by foster care, the adverse socio-emotional consequences appear not to be. If this is the parsimonious conclusion to be drawn from the data, it is still a very interesting and novel finding, but would require some adjustment in framing of the current manuscript.

It would be helpful to provide greater clarity about the pattern of associations within each group that produce the mediation effect described in the manuscript, and to consider whether these patterns would be better accounted for by the “dissociation” interpretation presented above.

Given that our exploration of the mediation analysis isn’t consistent with the alternate interpretations provided by the reviewer, we have left our primary interpretation of the findings in the paper. However, the Reviewer raises a very interesting alternate explanation of the difference in intervention effects between behavioral ratings and task performance. Our original interpretation of this difference was related to differences in measurement for behavioral tasks vs. teacher and parent reports of symptoms and social skills, as well as equifinality for symptoms of depression and social skill deficits. With regard to measurement, it is likely that symptoms of depression and social skills are less precisely measured than task performance. In addition, depression and social skills are influenced by a wider range of risk factors than those examined here. Thus we would see an intervention effect on our more carefully and precisely defined abilities (SRT, MID) but not on these more general indicators of functioning. The interpretation provided here by the reviewer is a potential alternate explanation that we now additionally include in the discussion section.

Reviewer #2 (Remarks to the Author):

This is a truly remarkable study, the challenges of which should not be understated. The randomization is a considerable strength. I think the data are of considerable interest to a wide readership, although probably not for the reasons stated by the authors. The comments below expand on these issues, but in brief, the study is not a direct examination of the "causal" effect of extreme social deprivation on later cognitive-emotional function, but rather a study of the effects of an intervention designed to provide a wide range of experience denied in early life.

I am somewhat puzzled why the authors have focused more on the intention of defining the effects of early social deprivation and less so on the effects (or lack of) the intervention? Also, it is sobering that despite the rather robust effects of the foster care treatment on experimental tasks, there was apparently little effect on emotional well being or social function, at least as rated by teachers/caretakers. Does this speak to the issue of reversibility?

The reviewer makes an important point that we did not randomly expose children to institutionalization and we have tried to be more explicit about this in the manuscript. When we refer to causal effects, we are referring to the intervention, which involved removing children from a deprived environment. In that sense, the intervention reflects randomization to prolonged institutional care versus being removed from a deprived early environment. In the paper, we have made this logic more clear and conceptualize intervention effects as isolating the impact of prolonged exposure to institutionalization.

A similar, though slightly different point, was raised by Reviewer 1, who asked us to examine the duration of institutionalization in addition to group assignment. In accordance with these suggestions, we ran additional analyses using percent time in the institution at age 12 as the primary IV. Average percent time spent in an institution by age 12 is 15% (9.3 SD) for the foster care group and is 43.9% (28.4% SD) for the care as usual group. As noted above, the pattern of findings is quite similar to those using group assignment when we focus on duration of institutional care. We provide these additional analyses in the revised manuscript. Given that the findings from these

analyses are consistent with findings from the intent to treat approach, we continue to interpret our findings as reflecting the impact of prolonged institutional exposure on cognitive function.

We agree that the fact that there is a main effect of institutionalization on depression and social skills, but not an intervention effect, is consistent with the possibility that the effects of early exposure to extreme deprivation (between 0-36 months) on depression and social skills may not be reversible. Importantly, we hesitate to interpret these findings too strongly because just as we didn't randomly assign children to institutionalization, we did not randomly assign children to grow up in families from birth; thus the ever institutionalized/never institutionalized differences can't be interpreted as causal.

I think both the introduction and especially the Discussion are over-written, and could be shortened considerably, with an emphasis on the more issues noted above.

Consistent with this reviewer's suggestion we have shortened both the discussion and introduction while also adding a focus on differences between FCG/CAUG and the NIG (ever institutionalized vs. never institutionalized analysis), as suggested above.

Finally, I think the experimental tests, while appropriate, do not inform on the specific neural processes at play.

We agree that we can not conclude what neural processes are supporting these cognitive and behavioral findings, we have removed references to neural function from the discussion which has shortened it, consistent with the above suggestion.

In sum, I would very much suggest a considerable re-writing that respects the actual design of the study (an intervention study that only indirectly informs on the impact of social deprivation) and the actual outcomes, or lack thereof, the intervention on socio-emotional function. The intense deprivation associated with institutionalized rearing is accompanied by multiple effects not likely reversed. For example, I recall data on growth stunting in these children as well as altered pubertal timing. Is there data on the general health of the groups (e.g., infection rates. Etc)? I think a more focused discussion of the reversible and, apparently, non-reversible effects is of relevance in relation to the multiple features of the

institutional setting. Moreover, if I understand, the placement of children in these institutions was non-random, thus imply adversity at earlier periods of development, including the prenatal period. Interestingly, Pearson et al (JAMA Psychiatry 2013) suggest that the transmission of risk from depressed mother to offspring is primarily during prenatal development – could this explain the negative findings with depressive symptoms? I believe the paper could broaden what is currently a rather narrow focus to discuss the negative findings on depressive symptoms and social skills.

We thank the reviewer for these comments and pointing us to this paper. In response to this and comments by Reviewer one have now expanded our discussion of the effects of early deprivation that can and cannot be reversed by random assignment into a family environment in the discussion.

Comments

p. 2: The authors claim that “The absence of a caregiver results not only in atypical social and emotional development, ... but also marked differences in cognitive development.⁸” This statement implies causality and seems at odds with the gist of the abstract, which rightly notes the limitation of previous studies as correlational.

We have changed this to ‘is associated with’.

p. 3: The phrase “other forms of early-life adversity such as early life stress broadly defined” is rather meaningless – why not simply “other forms of early life adversity”? Presumably early life adversity and early life stress are somewhat synonymous. I am also unclear as to what is meant by “disrupted parenting” – this should be more precise, such as in the other examples.

The author of this study (Hanson, Albert, et al., 2015) describe the exposures in their sample as ‘early life stress.’ Early life stress posits a specific mechanism (stress) whereas early life adversity does not. Disrupted parenting was alternately described as maltreatment in this study so we have used that term instead in the revision with a reference to studies about abuse.

p. 3: I am also confused about the source of the evidence for the statement “These disruptions in reward processing have been associated with increased risk for depression^{22,23} and social skills deficits²⁰ following exposure to early adversity. ” Is there mediational analyses? Deficits in reward processing are indeed associated with a risk for depression, but this is not unique to individuals experiencing early life adversity.

These are mediation analyses.

p. 3: RDoC is a working model, not a source of evidence.

Thanks for noting this, we have changed our wording in this section.

p. 4: I think the authors are actually doing a disservice to their work. The model does not actually address the issue the “causal relationship between early life adversity with reward and implicit learning” rather it speaks to a more specific form of adversity, that of the lack of age-appropriate social interactions with adults. It does not address the issue of ‘early life adversity’, which can take many forms. I think this is a critical point as this literature tends to very wrongfully lump multiple, very diverse forms of experience as “adversity”, which does a real disservice to the developmental science. Finally, an experiment that address the causal role of an adversity would randomly assign individual to a condition of that adversity.

We are whole heartedly in agreement that the field suffers from an abundance of work on poorly defined ‘adversity’ and we think the specific contribution of this paper is to better understanding prolonged exposure to severe deprivation in early life. We have endeavored to limit our introduction and interpretation to this concept. We have revised throughout to make it clear that we are specifically referring to early-life deprivation. When we retain the more global term adversity, we do so because that is how the original research we are citing conceptualized and measured the exposure.

It seems from the comments here that there is some disagreement about if we can conceptualize the randomization as speaking to the causal impact of deprivation on cognitive function. We believe we can. Because the children who were randomized out of the institutions spent overall considerably less time in the institutions by age 12 (see

Table 1), the randomization did in fact change the degree of exposure to extreme deprivation between children who were and were not randomized to foster care. Additionally, as we outline in our response to Reviewer 1, percent time in the institution, which could be conceptualized as an index of 'amount of exposure to deprivation', has an identical pattern of results with our outcomes as does our intent to treat analysis. We have added these additional analyses to the revised manuscript.

p. 4: Regarding the "An age-matched sample of 72 community-reared children was recruited from pediatric clinics..." were these children seen for specific disorders/treatments, or routine pediatric examinations? I think this needs details. Also, was there analysis of the home environments or parent-offspring interactions in the foster care settings? This is actually the intervention, and yet is poorly described.

The control group in this study is comprised of children who were born in Bucharest at the same time as participants in the randomized control trial at similar hospitals but who were raised by families in the community and were not placed into institutions. They were seen at the clinics for routine care and were healthy when enrolled into the study. Because the NIG were not randomly assigned in any fashion, we can not use comparisons to this group for anything other than benchmarking changes from an atypical trajectory observed in the FCG. Differences between the FCG and NIG or the ever and never institutionalized groups are associative (i.e., not based on random assignment) as most of the work in this field is.

p. 4: Were the 6 institutions of comparable care (sic)? Are there systematic differences between children placed in institutions and those that remain within their families? This bears on the later FCG and CAUG to NIG comparisons.

The 6 institutions were comparable, and no differences between children from these 6 institutions have been observed in this sample (see Zeanah et al., 2003 for greater details). Because of an historical reliance on institutional care for abandoned children in Romania, foster care was almost nonexistent in Bucharest at the time the study began and it was culturally normative to place children in institutions if one was too poor or otherwise unable to care for children. In addition, the political climate that preceded this study included structural barriers to contraception, including outlawing

contraception for a time and laws which actively encouraged child-bearing but did not provide social safety nets for parents. At study onset all participants were carefully screened for evidence of prenatal substance exposure and exposure to abuse and have been subsequently screened for known genetic diseases. Please see Zeanah et al., 2003 for greater details on the study design and historical context of institutional rearing in Romania at the time the study began.

p. 4: Can the authors elaborate on "care as usual (prolonged institutional care; CAUG)."? This seems to be the critical comparison group.

These were the children who were included in the study when they were infants and have been followed longitudinally until age 12; when they were between 6-33 years of age they were randomly assigned to stay in the institution, or to care as usual (CAUG), rather than being removed from institutional care. We included these participants in the study, we followed them longitudinally, but we did not interfere with their care in the Romanian system. As a result these children spent substantially more time in institutional care relative to those in the foster care group who were randomly assigned to be removed from institutional care and placed into high-quality foster care with a family between 6-33 months of age. As can be observed in Table 1, on average children in the CAUG spent 44% of their life in an institution whereas those in the FCG spent only 15% of their lives in an institution. This difference is the result of random assignment and is the difference under investigation here.

p. 7: I do not really understand how group differences are interpreted as "the effect of early caregiving environment". This is not what was manipulated and seems counter to the design, which is an intervention study. Why would group differences not be associated with the placement in foster care? They were not randomly assigned to adverse conditions?

We have changed this language to 'prolonged institutional care' and 'continued exposure to a deprived environment after infancy,' which better describe the difference in experience between the CAUG and FCG. The intervention was specifically designed to remove children from a deprived early environment, and thus the differences between the groups reflect differences in the duration of time spent in that deprived early environment.

"Consistent with our hypotheses, we observed a significant indirect effect of randomized placement into foster care on parent (95% bias corrected confidence interval (CI): [.003, .072]) and teacher (CI: [.004, .042]) report of depressive symptoms through reward learning." As noted below, reward sensitivity is a defining feature of depressive states, so this finding is not surprising. This link would be strengthened by reference to high-risk populations (and see below). Are there data on drug use in these samples (e.g., cigarette use)? I am also somewhat surprised by the focus on depression and an additional assessment of anxiety-related symptoms, which normally show an earlier onset.

While anhedonia is a defining feature of depression, response to reward on a task like this is not. Reward tasks like this are not used for diagnosis nor are they used to index clinically meaningful change in treatment settings. While it is consistent with modern understandings of depression to examine changes in reward sensitivity as a mediator of the impact that early-life deprivation has on symptoms of depression, it is not a given that this mediation would be significant. Indeed, depression, like all mental disorders is multi-determined and heterogeneous. It is possible to get a diagnosis of depression without endorsing symptoms of anhedonia. The fact that we do observe an effect of randomization to foster care on reward sensitivity but not on depression is consistent with, and likely related to, this heterogeneity.

p. 12: The authors "examined whether pattern learning on the SRT significantly mediated the effect of intervention (CAUG vs. FCG) on depressive symptoms." But on p. 10 the authors state that "There were no significant differences in symptoms of depression between children in the FCG and CAUG (p 's .16, .40)...", suggesting no intervention effect.

That is correct, there was not a significant effect of randomization on symptoms of depression or quality of peer relationships. Identifying mediations in the absence of a significant c path is common practice (Hayes, 2009; Hayes & Matthes, 2009; Preacher & Hayes, 2004, 2008). It is possible that mediators can mask significant associations between the IV and DV so that only an indirect effect of the predictor through the mediatory is observable, as we see here.

p. 12: I think the lead statement in the Discussion, for reasons also noted above, overstates

(misstates) the results "In the current study we demonstrate that early institutional rearing causes deficits in reward and implicit learning observable in early adolescence and that these differences explain later impairments in both depression and social skills. ". The study directly examined the effects of an intervention, that included dimensions associated with institutionalization – this is well off the claim of evidence for a causal relation between institutionalization and the performance differences in the specific tasks.

We have tempered this language in the revision. The reviewer is correct that we did not randomly assign children to institutionalization. We did, however, randomize them to be removed from that deprived environment. The intervention effects thus reflect meaningful differences in the duration of exposure to a deprived early environment. We have made sure this language is more specific in the revision.

Moreover, there were no differences in depressive symptoms between the FCG and CUAG groups, thus contradicting this statement.

We agree, we have changed this language through out the discussion.

Likewise on p. 10 the authors acknowledge that there were "no significant differences between children in the FCG and CAUG in social skills." The merits of the mediational models, which appears driven with the addition of the NIG group, is that the deficits in the reward and implicit learning tasks appear to be associated with the behavioral problems – however, this remains based on correlational models.

We disagree with this interpretation based on our running of subsequent models examining only the CAUG and FCG and using percent time in the institution as a IV, which we have included in the revised manuscript. These analyses demonstrate clearly differences related to randomization. In addition, Reviewer 1 provided several potential alternate explanations of our findings based on potential patterns of associations within each group (CAUG, FCG, NIG). As you can see in our response to that reviewer, these potential patterns were not observed in our data and our primary interpretation of our results is sound.

p. 12: I do not see how this is a “, implicit motor learning” – what is the motor learning component?

The SRT task is generally described as an implicit motor learning task to distinguish it from other types of implicit learning tasks (e.g., an implicit auditory learning task). Here the motor component is the pattern of button presses. Initially, these responses are slow and inaccurate before the pattern is learned, but as the same pattern appears again and again the motor output becomes more automatized and thus, faster and more accurate. A real-world example of this kind of motor learning would be learning to play a song on a piano or another musical instrument, or learning how to type a particular word. Here, the speeded response on patterned blocks reflects learning of the underlying pattern, whereas on un-patterned blocks motor responses simply reflect the baseline reaction time for pressing keys (as there is no underlying pattern in the numbers presented). Importantly, it is widely understood that implicit learning in a variety of modalities (e.g., visual, auditory, motor) is subserved by a similar set of subcortical and cortical regions (Johnson, Turk-Browne, & Goldberg, 2016; Schapiro, Turk-Browne, Botvinick, & Norman, 2017; Schlichting, Guarino, Schapiro, Turk-Browne, & Preston, 2016). The serial reaction time task is a standard task for studying implicit motor learning (Robertson, 2007; Romano, Howard, & Howard, 2010; van der Graaf, Maguire, Leenders, & de Jong, 2006).

p. 13: I think comparative anatomists would argue against the notion that “neural structure and function which underlies associative learning is conserved across species.” – the incentive components, perhaps, but the neocortical components – not so much.

We agree, we have modified this claim in the paper.

That aside, the authors have used appropriate tests for forms of learning – the nature of the impaired performance, however, remains to be determined. There are, as I am sure they appreciate, attentional mechanisms, amongst other functions, that contribute to performance. And the absence of a negative control (i.e., a learning task on which the FCG/CAUG and NIG groups do not differ) limits interpretation.

Indeed, it is the case that we see many deficits in children exposed to institutionalization, including those in executive function. Interestingly, in other analyses we have demonstrated that these do not respond to foster care intervention (i.e., we observe only an ever institutionalized – never institutionalized difference, not a randomization difference as we see here; Tibu et al., 2015, 2016). It is possible that attention, or as the reviewer points out in the below comment, a general impairment in motor function, interacts with task demands to influence our results. However, given that our manipulations were designed to examine the impact of randomization to foster care on reward sensitivity and implicit motor learning, we feel it is most parsimonious to interpret them in this way.

p. 14: There is a circularity in statements that associated reward learning to depression...anhedonia is a defining feature, perhaps the defining feature, of depression. Hence by definition reward processing is associated with depression. The studies of Ian Gotlib with high-risk subjects may help address this point. Unlike the current paper, the model cited shows specific processing deficits and neural circuits derived from fMRI analyses.

We have tried to modify these statements to make them less circular. However, altered reward processing per se is not synonymous with depression. As noted above, anhedonia is not required for a diagnosis of depression and there is enormous heterogeneity in depression that likely reflects different underlying sub-groups (though clear delineation of such sub-types remains an ongoing challenge for the field and beyond the scope of the current paper).

With regard to circularity more generally, there is documented evidence that disruptions in reward processing at both behavioral and neurobiological levels *precede* the onset of depression – Gotlib’s work is central here, and we believe this is what the reviewer is referring to. Specifically, blunted striatal response during reward anticipation and receipt has been observed in adolescent females who are at-risk for depression due to strong family history but who have not yet experienced a depressive episode (Colich et al., 2017; Gotlib & Joormann, 2010). Longitudinal studies also demonstrate that blunted behavioral response to reward, a pattern similar to that observed in our findings, predicts the subsequent onset of depression in

children(Forbes, Shaw, & Dahl, 2007). These findings held even after controls for prior psychopathology. Thus, existing evidence is consistent with a developmental pathway whereby adversity exposure influences neural development and reward processing, which in turn influence risk of depression.

We have included this argument more clearly in the revised paper to make it clear how existing work aligns with our conceptual model and does not align with a model wherein reward processing deficits are simply reflecting the presence of depression.

p. 15: "We observe that reward learning was entirely remediated by random assignment to a family environment between 6 and 33 months of age; we observe no differences in performance between children who were randomized to foster care or who grew up in a family environment from birth on this task (Figure 2)." This is indeed perhaps the strongest finding. One issue is that of decomposing the performance deficits. Is it relevant that the groups differed very significantly on the 'random' condition in the implicit learning task? Would I be correct to assume that this would seem to be a purer test of memory – might this speak to the issue of specificity? What does it say about the children in general?

The fact that children differed in performance on the random condition indeed suggests a general deficit in speeded motor performance or speeded processing more generally. Because general deficits like this are common in children exposed to severe adversity we always control for baseline conditions such as these to isolate the effect of implicit motor learning or sensitivity to reward relative to general motor function.

One the same point, I am somewhat puzzled by the term "reward learning"? What is actually learned here? This test appears to examine reward-sensitive performance, which is not trivial. But I do not see the actual "learning" component?

We agree that the learning component of this task was over-emphasized in the current version of the manuscript. In accordance with this comment and the suggestions by Reviewer 1 we have edited references to the MID and now describe them as reflecting reward sensitivity.

Reviewer #3 (Remarks to the Author):

This manuscript examined associations between early deprivation, reward learning and implicit learning, and depressive symptoms and social skills. It is the first experimental (non-correlational) study to report that associative learning mediates associations between early deprivation and adolescent behavioral outcomes. Researchers in the field have speculated that early stress influences mental health and social skills via associative learning, and it is exciting to see preliminary experimental evidence for this relationship. The statistical analyses appear to be sound and the manuscript is very well-written. I only have a few concerns, which are listed below.

The measurement of depression (7 items) is limited. It is good that the authors obtained both parent and teacher reports, but an additional child report and a more detailed measure like the CDI would make the findings more robust. The limited measurement of depressive symptoms should be discussed.

We agree and we have now listed this as a limitation in the discussion. We hope that in future follow-up we will be able to obtain self-report of symptoms of psychopathology.

For social skills, the items used seem to measure primarily peer relationships, but not broader social skills. The authors might consider using the term "peer relationships" to describe this construct.

We agree, and have changed this wording in the manuscript.

It would be useful to see a graph or table showing the distributions of scores for depression and social skills in each group.

We have included this information in the current version of the manuscript, please see Table 1.

Was pubertal development measured in this study? Puberty is often more advanced in children exposed to early stress, and is also related to depression and reward learning.

Puberty was measured in this study using the Petersen scale, and as we have reported elsewhere, differs by exposure to institutionalization. However, we did not observe any significant associations between puberty and either MID ($b = -.012$, $t = -.123$, $p = .902$) or SRT ($b = .072$, $t = 1.245$, $p = .22$) performance. These null associations with task performance were unchanged when we removed age from the analysis. It maybe that within the narrow age range we are assessing (all children were 12 or 13 years of age) we do not have enough variance to observe previously reported associations between puberty and reward.

Did any children fail to perform above chance on the MID and SRT, and were they eliminated from data analysis?

One child was unable to complete the task due to fatigue, we report this in the methods. We did not eliminate any children from the analysis for task performance for several reasons. First children's performance was monitored very closely by study staff so that no child was completely disengaged from task performance. This is corroborated by main effects of task manipulations in low performing participants. For example, even if we only examined children for whom there was at least one instance of chance task performance we see an effect of monetary incentive on task performance ($F_{(3,36)} = 5.32$, $p = .004$) and a non-significant effect of pattern on task performance ($F_{(2,19)} = 1.92$, $p = .18$). This suggests that even for subjects who had low accuracy on the task, performance varies according to task conditions as we would expect.

Additionally, even though we selected simple tasks with used in populations much younger than these children, response time is generally slow for children in the previously institutionalized group, even when they are clearly trying hard on the task. This overall motor slowing is evidenced by a large main effect of group on average reaction time, regardless of other task demands (all p 's $< .003$ for the main effect of group on RT across conditions). This global slowing in reaction time is why we always examine *relative* performance (4 star vs. 0 star; patterned vs. random), eliminating a main effect of reaction time from driving our results. Finally, we decided not to exclude

children who performed less than chance on these tasks because excluding participants will violate intent-to-treat and disallow causal inference.

It should be noted in limitations that this study was not completely experimental, given the inclusion of the never-institutionalized group.

For comparisons where we were identifying potential causal effects we directly compare the FCG to the CUAG without including the NIG to deal with this potential problem.

REFERENCES

Cohen, J. M., Cohen, P., West, S., & Aiken, L. (2003). *Applied Multiple Regression/Correlation*

Analysis for the Behavioral Sciences. New Jersey: Lawrence Erlbaum Associates

Publishers.

Colich, N. L., Ho, T. C., Ellwood-Lowe, M. E., Foland-Ross, L. C., Sacchet, M. D., LeMoult, J. L.,

& Gotlib, I. H. (2017). Like mother like daughter: putamen activation as a mechanism

underlying intergenerational risk for depression. *Social Cognitive and Affective*

Neuroscience, *12*(9), 1480–1489. <https://doi.org/10.1093/scan/nsx073>

Forbes, E. E., Shaw, D. S., & Dahl, R. E. (2007). Alterations in reward-related decision making

in boys with recent and future depression. *Biological Psychiatry*, *61*(5), 633–639.

<https://doi.org/10.1016/j.biopsych.2006.05.026>

Gotlib, I. H., & Joormann, J. (2010). Cognition and Depression: Current Status and Future Directions. *Annual Review of Clinical Psychology, 6*, 285–312.

<https://doi.org/10.1146/annurev.clinpsy.121208.131305>

Hayes, A. F. (2009). Beyond Baron and Kenny: Statistical Mediation Analysis in the New Millennium. *Communication Monographs, 76*(4), 408–420.

Hayes, A. F., & Matthes, J. (2009). Computational procedures for probing interactions in OLS and logistic regression: SPSS and SAS implementations. *Behavior Research Methods, 41*(3), 924–936. <https://doi.org/10.3758/BRM.41.3.924>

Johnson, M. A., Turk-Browne, N. B., & Goldberg, A. E. (2016). Neural systems involved in processing novel linguistic constructions and their visual referents. *Language, Cognition and Neuroscience, 31*(1), 129–144.

<https://doi.org/10.1080/23273798.2015.1055280>

Nelson, C. A., Zeanah, C. H., Fox, N. A., Marshall, P. J., Smyke, A. T., & Guthrie, D. (2007). Cognitive recovery in socially deprived young children: the Bucharest Early Intervention Project. *Science (New York, N.Y.), 318*(5858), 1937–1940.

<https://doi.org/10.1126/science.1143921>

Preacher, K. J., & Hayes, A. F. (2004). SPSS and SAS procedures for estimating indirect effects in simple mediation models. *Behavior Research Methods, Instruments, & Computers: A Journal of the Psychonomic Society, Inc*, *36*(4), 717–731.

Preacher, K. J., & Hayes, A. F. (2008). Asymptotic and resampling strategies for assessing and comparing indirect effects in multiple mediator models. *Behavior Research Methods*, *40*(3), 879–891.

Robertson, E. M. (2007). The Serial Reaction Time Task: Implicit Motor Skill Learning? *The Journal of Neuroscience*, *27*(38), 10073–10075.

<https://doi.org/10.1523/JNEUROSCI.2747-07.2007>

Romano, J. C., Howard, J. H., & Howard, D. V. (2010). One-Year Retention of General and Sequence-Specific Skills in a Probabilistic, Serial Reaction Time Task. *Memory (Hove, England)*, *18*(4), 427–441. <https://doi.org/10.1080/09658211003742680>

Schapiro, A. C., Turk-Browne, N. B., Botvinick, M. M., & Norman, K. A. (2017). Complementary learning systems within the hippocampus: a neural network modelling approach to reconciling episodic memory with statistical learning. *Philosophical Transactions of the Royal Society of London. Series B, Biological Sciences*, *372*(1711).

<https://doi.org/10.1098/rstb.2016.0049>

Schlichting, M. L., Guarino, K. F., Schapiro, A. C., Turk-Browne, N. B., & Preston, A. R. (2016).

Hippocampal Structure Predicts Statistical Learning and Associative Inference Abilities during Development. *Journal of Cognitive Neuroscience*, *29*(1), 37–51.

https://doi.org/10.1162/jocn_a_01028

Tibu, F., Sheridan, M. A., McLaughlin, K. A., Nelson, C. A., Fox, N. A., & Zeanah, C. H. (2015).

Disruptions of working memory and inhibition mediate the association between exposure to institutionalization and symptoms of attention deficit hyperactivity disorder. *Psychological Medicine*, 1–13. <https://doi.org/10.1017/S0033291715002020>

Tibu, F., Sheridan, M. A., McLaughlin, K. A., Nelson, C. A., Fox, N. A., & Zeanah, C. H. (2016).

Disruptions of working memory and inhibition mediate the association between exposure to institutionalization and symptoms of attention deficit hyperactivity disorder. *Psychological Medicine*, *46*(3), 529–541.

<https://doi.org/10.1017/S0033291715002020>

van der Graaf, F. H. C. E., Maguire, R. P., Leenders, K. L., & de Jong, B. M. (2006). Cerebral

activation related to implicit sequence learning in a Double Serial Reaction Time task.

Brain Research, *1081*(1), 179–190. <https://doi.org/10.1016/j.brainres.2006.01.103>

Winter, W., & Sheridan, M. (2014). Previous reward decreases errors of commission on later "No-Go" trials in children 4 to 12 years of age: evidence for a context monitoring account. *Developmental Science*. <https://doi.org/10.1111/desc.12168>

Zeanah, C. H., Nelson, C. A., Fox, N. A., Smyke, A. T., Marshall, P., Parker, S. W., & Koga, S. (2003). Designing research to study the effects of institutionalization on brain and behavioral development: the Bucharest Early Intervention Project. *Development and Psychopathology*, *15*(4), 885–907.

Reviewers' comments:

Reviewer #1 (Remarks to the Author):

I appreciate the efforts that the authors put into this revision. The additional description of the task methods and the study population has improved the clarity of the study design. I also appreciate the inclusion in the discussion of the points raised about MID in relation to reward learning, and the specificity of the effects of foster care observed in this study to cognitive (versus socio-emotional) processing .

The authors are correct that I misunderstood the analytical approach in their regression analyses. I now realize that they are using an effect coding strategy (to define hypothesized statistical contrasts of interest across categorical variables) and this fully addresses my previous confusion about the seeming discrepancy in the directionality of the effects, as how the coding pertains to the mediation analyses. The inclusion of the control analysis (contrasting FCG and CAUG to NIG) is a helpful addition.

I have no remaining concerns about the analyses or their interpretation. This round of revision has augmented the strength and clarity of an already-strong manuscript. I recommend acceptance of this manuscript, which makes an important and novel scientific contribution.

Reviewer #2 (Remarks to the Author):

I think the authors have responded convincingly to many points in the preparation of the revised version of the paper. Issues associated with interpretation of the task, details of the environmental conditions for the children and the relation between reward processing and depressive symptoms have been well considered in the revision. However, I remain of the opinion that the authors have overstated the strength of the relationship between the social deprivation and specific outcome measures (i.e., "The use of random assignment allows assessment of the causal relationship between early life deprivation with reward and implicit learning."

I feel the section below more accurately reflects the design of the study: "We use this experimental data to determine whether randomized placement into family care mitigates the impact of early institutionalization on associative learning and whether improvements in associative learning are a mechanism explaining group differences in mental health and psychosocial outcomes later in development." This seems a very reasonable statement of the objectives of the study. The authors refer to the placement as "randomization to a high-quality foster care intervention".

"The use of random assignment allows assessment of the causal relationship between early life deprivation with reward and implicit learning." I disagree. It best allows for a statement about the causal relation between the intervention and the outcome. The authors "

randomization out of a psychosocially and cognitively deprived institutional environment " and into a "a high-quality foster care intervention ". The conclusion regarding a "causal" relation between social deprivation and outcome ignores the placement into an intervention program.

I am somewhat puzzled that the authors "agree that the fact that there is a main effect of institutionalization on depression and social skills, but not an intervention effect, is consistent with the possibility that the effects of early exposure to extreme deprivation (between 0-36 months) on depression and social skills may not reversible. Importantly, we hesitate to interpret these findings too strongly...". And yet the study was actually designed to examine the effects of an intervention? Nevertheless the authors are rather ambitious in defining a causal effect of social deprivation, for the study was not actually designed (ie., random assignment to conditions of social deprivation from a common background). This seems counter-intuitive. In sum, the authors found little difference between the CAUG and FCG groups on measures of socio-emotional function. This is an important, if rather discouraging finding.

I think the authors have constructively responded with the time in deprivation analyses. This is a heroic and remarkable study. The data will be of considerable interest to researchers at the interface of neuroscience and child psychopathology. The findings could be interpreted as "being consistent with a causal relationship" - but not clearly defining such an effect. I think this wording might provide a realistic interpretation of the findings aligned to the design and statistical analyses.

Reviewer #3 (Remarks to the Author):

Thank you for the invitation to review this revision. The revised manuscript is much improved from the previous version, and the authors addressed all of my previous questions and concerns. However, I do have a few more suggestions that would improve the manuscript, outlined below:

Introduction:

p. 3, 2nd paragraph: The authors should note the probabilistic learning task in Hanson et al. was substantially different than the implicit learning task used in Eigsti et al. I would not consider the Hanson et al. task a pure implicit learning task, it is more cognitively complex than the Eigsti task. The authors compare these two studies again on p. 16, 2nd paragraph.

In general, there seems to be much more background on reward learning than implicit learning in these stress-exposed populations. The authors could provide a better rationale for why they hypothesized implicit learning would be affected by institutionalization.

Results:

Table 1: indicate which groups were statistically different, perhaps using subscripts

It would be helpful to provide figures depicting the significant mediation analyses. If the # of figures is a concern, these could be combined with the figures for overall group differences on each task.

Discussion

p. 17, first full paragraph seems incomplete: WHY might we observe that reward learning was entirely remediated by random assignment whereas implicit learning was not? Some inference would add depth to this paragraph.

Reviewers' comments:

Reviewer #1 (Remarks to the Author):

I appreciate the efforts that the authors put into this revision. The additional description of the task methods and the study population has improved the clarity of the study design. I also appreciate the inclusion in the discussion of the points raised about MID in relation to reward learning, and the specificity of the effects of foster care observed in this study to cognitive (versus socio-emotional) processing.

The authors are correct that I misunderstood the analytical approach in their regression analyses. I now realize that they are using an effect coding strategy (to define hypothesized statistical contrasts of interest across categorical variables) and this fully addresses my previous confusion about the seeming discrepancy in the directionality of the effects, as how the coding pertains to the mediation analyses. The inclusion of the control analysis (contrasting FCG and CAUG to NIG) is a helpful addition.

I have no remaining concerns about the analyses or their interpretation. This round of revision has augmented the strength and clarity of an already-strong manuscript. I recommend acceptance of this manuscript, which makes an important and novel scientific contribution.

We thank the Reviewer for this thoughtful and fair review, we are happy to hear we have addressed their concerns.

Reviewer #2 (Remarks to the Author):

I think the authors have responded convincingly to many points in the preparation of the revised version of the paper. Issues associated with interpretation of the task, details of the environmental conditions for the children and the relation between reward processing and depressive symptoms have been well considered in the revision. However, I remain of the opinion that the authors have overstated the strength of the relationship between the social deprivation and specific outcome measures (i.e., "The use of random assignment allows assessment of the causal relationship between early life deprivation with reward and implicit learning."

We take this reviewers point and we agree that the language below best describes the intervention and what we can learn from this data. In the manuscript we have carefully looked for every instance where we describe the the intent to treat analysis and have tried to re-state the meaning of these findings using the language suggested here. (e.g., last paragraph of page 4 now states "The use of random assignment allows assessment of the mitigating impact of foster care intervention on the association between early life deprivation and reward and implicit learning." The changes can be found on pages 4, 9, 15, 17, and 18.

I feel the section below more accurately reflects the design of the study: "We use this experimental data to determine whether randomized placement into family care mitigates the impact of early institutionalization on associative learning and whether improvements in associative learning are a mechanism explaining group differences in mental health and psychosocial outcomes later in development." This seems a very reasonable statement of the objectives of the study. The authors refer to the placement as "randomization to a high-quality

foster care intervention".

"The use of random assignment allows assessment of the causal relationship between early life deprivation with reward and implicit learning." I disagree. It best allows for a statement about the causal relation between the intervention and the outcome. The authors " randomization out of a psychosocially and cognitively deprived institutional environment " and into a "a high-quality foster care intervention ". The conclusion regarding a "causal" relation between social deprivation and outcome ignores the placement into an intervention program.

Here and elsewhere we have tried to modify our language to emphasize the impact of the intervention (into high quality foster care) consistent with this reviewer's recommendation. See above and in tracked changes to identify where these changes have been made.

I am somewhat puzzled that the authors "agree that the fact that there is a main effect of institutionalization on depression and social skills, but not an intervention effect, is consistent with the possibility that the effects of early exposure to extreme deprivation (between 0-36 months) on depression and social skills may not reversible. Importantly, we hesitate to interpret these findings too strongly...". And yet the study was actually designed to examine the effects of an intervention?

We're sorry if this point was confusing, we are cautious in interpreting main effects of institutionalization (i.e., comparisons between children raised in families since birth and those with any exposure to institutionalization) because those effects could be caused by unmeasured third variables. Additionally, as it is a null result, interpretation of the lack of an intervention effect is hampered, by power issues. So on the one hand, these findings are definitely consistent with the reviewer's interpretation, but there are other possible interpretations given these limitations. Thus, we interpret them cautiously.

Nevertheless the authors are rather ambitious in defining a causal effect of social deprivation, for the study was not actually designed (ie., random assignment to conditions of social deprivation from a common background). This seems counter-intuitive. In sum, the authors found little difference between the CAUG and FCG groups on measures of socio-emotional function. This is an important, if rather discouraging finding.

We agree. We had enough power to identify intervention differences in reward responsivity and implicit learning and it is certainly discouraging that, in the light of those findings, we did not observe differences in social-emotional function. We have tried to highlight these observations, along with this potential interpretation in this version of the manuscript on page 17.

I think the authors have constructively responded with the time in deprivation analyses. This is a heroic and remarkable study. The data will be of considerable interest to researchers at the interface of neuroscience and child psychopathology. The findings could be interpreted as "being consistent with a causal relationship" - but not clearly defining such an effect. I think this wording might provide a realistic interpretation of the findings aligned to the design and statistical analyses.

We agree and we have changed our language accordingly as we mention above.

Reviewer #3 (Remarks to the Author):

Thank you for the invitation to review this revision. The revised manuscript is much improved from the previous version, and the authors addressed all of my previous questions and concerns. However, I do have a few more suggestions that would improve the manuscript, outlined below:

Introduction:

p. 3, 2nd paragraph: The authors should note the probabilistic learning task in Hanson et al. was substantially different than the implicit learning task used in Eigsti et al. I would not consider the Hanson et al. task a pure implicit learning task, it is more cognitively complex than the Eigsti task. The authors compare these two studies again on p. 16, 2nd paragraph.

We take the reviewers point, we now have included information about the limitation of these comparisons in this paragraph (pages 3 and 16).

In general, there seems to be much more background on reward learning than implicit learning in these stress-exposed populations. The authors could provide a better rationale for why they hypothesized implicit learning would be affected by institutionalization.

The rationale was two-fold. First, from a theoretical perspective reward learning relies on the same neural circuits as implicit learning and is a related form of learning. It seemed likely that if these circuits were disrupted by a multi-dimensional adversity such as institutionalization (as evidenced by previous findings using reward), other aspects of learning would be disrupted, but very few studies had tested this idea. Second, there was one study, by one of the authors of this manuscript, which emphasized the link between associative learning early in life and later social processes. This link also has robust theoretical grounding but also has seldom been tested. This seemed like an apt opportunity to test these ideas (please see page 3, final paragraph) .

Results:

Table 1: indicate which groups were statistically different, perhaps using subscripts

We have made this change (see Table 1).

It would be helpful to provide figures depicting the significant mediation analyses. If the # of figures is a concern, these could be combined with the figures for overall group differences on each task.

We now include figures depicting the mediation analyses (Figure 4).

Discussion

p. 17, first full paragraph seems incomplete: WHY might we observe that reward learning was entirely remediated by random assignment whereas implicit learning was not? Some inference would add depth to this paragraph.

We have now tried to include some of our theorizing about the source of this difference (page 18 “Interpreting the difference in the effect of foster care between these two interventions is complicated by the fact that they are measured in different tasks. However, it

maybe that where reward responsivity - by definition a more motivated form of associative learning - can engage multiple neural systems to support task performance, the SRT does not. Thus, group differences in SRT performance may better capture underlying group differences in basic associative learning processes.”)

REVIEWERS' COMMENTS:

Reviewer #3 (Remarks to the Author):

This revised manuscript addresses all my concerns, and makes a significant contribution to the field as the first paper to systematically examine links between early deprivation, associative learning, and social adjustment outcomes. I recommend this manuscript for publication in Nature Communications.